# Evaluation on Air Purifier’s Performance in Reducing the Concentration of Fine Particulate Matter for Occupants according to its Operation Methods

**DOI:** 10.3390/ijerph17155561

**Published:** 2020-08-01

**Authors:** Hyungyu Park, Seonghyun Park, Janghoo Seo

**Affiliations:** 1Department of Architecture, Graduated School, Kookmin University, Seoul 02707, Korea; sang911008@kookmin.ac.kr; 2Department of Industry-Academic Cooperation Foundation, Kookmin University, Seoul 02707, Korea; marine86@kookmin.ac.kr; 3School of Architecture, Kookmin University, Seoul 02707, Korea

**Keywords:** particulate matters (PM), air purifier, experiment, real-time monitoring unit, transfer unit, occupant, breathing zone

## Abstract

Fine particulate matter entering the body through breathing cause serious damage to humans. In South Korea, filter-type air purifiers are used to eliminate indoor fine particulate matter, and there has been a broad range of studies on the spread of fine particulate matter and air purifiers. However, earlier studies have not evaluated an operating method of air purifiers considering the inflow of fine particulate matter into the body or reduction performance of the concentration of fine particulate matter. There is a limit to controlling the concentration of fine particulate matter of the overall space where an air purifier is fixed in one spot as the source of indoor fine particulate matter is varied. Accordingly, this study analyzed changes in the concentration of indoor fine particulate matter through an experiment according to the discharging method and location of a fixed air purifier considering the inflow route of fine particulate matter into the body and their harmfulness. The study evaluated the purifiers’ performance in reducing the concentration of fine particulate matter in the occupants’ breathing zone according to the operation method in which a movable air purifier responds to the movement of occupants. The results showed the concentration of fine particulate matter around the breathing zone of the occupants had decreased by about 51 μg/m^3^ compared to the surrounding concentration in terms of the operating method in which an air purifier tracks occupants in real-time, and a decrease of about 68 μg/m^3^ in terms of the operating method in which an air purifier controls the zone. On the other hand, a real-time occupant tracking method may face a threshold due to the moving path of an air purifier and changes in the number of occupants. A zone controlling method is deemed suitable as an operating method of a movable air purifier to reduce the concentration of fine particulate matter in the breathing zone of occupants.

## 1. Introduction

Recently, high levels of fine particulate matter are observed in the atmosphere of the East Asian regions, including South Korea, regardless of the season due to rapid economic growth and industrialization, causing adverse effects on the body [1,2]. The inflow of outdoor particulate matter into the indoor space is causing deterioration of the indoor air quality. According to the World Health Organization, 3.8 million people a year die prematurely from illness attributable to household air pollution [3]. Particulate matter is classified into PM_10_, which has particulate matter less than 10 μm and PM_2.5_, which has particulate matter less than 2.5 μm [4]. Particles, in general, are filtered using the cilia or respiratory tract and cannot enter the lung when they are under the body, but fine particles of small size can penetrate the lung and accumulate in the alveoli. Particles accumulated in the alveoli cause an inflammatory response and generate active oxygen, causing necrotized tissues. When particles accumulate in the bronchial tubes, they cause phlegm and cough as well as drying of the bronchial mucosa, which can easily allow penetration of germs and leave people with chronic lung conditions vulnerable to infectious diseases like pneumonia [5,6]. In particular, fine particulate matter less than 2.5 μm have been reported to cause diseases such as angina, stroke, and heart attack [7,8,9,10]. This indicates that fine and ultrafine particulate matter penetrating the body through breathing can cause serious damage to the body.

Filter-type air purifiers are used to control indoor fine particulate matter in South Korea, and there has been a broad range of studies on air purifiers and fine particulate matter. However, previous studies focused on the purification effects caused by the airflow rate and filter grade of air purifiers and have not evaluated the operating method of air purifiers that considers the inflow path of fine particulate matter into the body and the breathing zone of occupants or the performance in reducing the concentration of particulate matter [11,12,13,14,15,16,17,18,19]. As various sources cause indoor fine particulate matter [20], operating a fixed air purifier can reduce the average concentration of indoor fine particulate matter, but the reduction effect against fine particulate matter in local areas such as the breathing zone of occupants is inadequate. Contrarily, we thought that delivering purified air discharged from the air purifier to the breathing zone of occupants by changing the discharge angle of the air purifier and attaching a transfer unit can reduce the concentration of fine particulate matter in the air that occupants breathe regardless of the average concentration of particulate matter in indoor spaces.

Accordingly, this study considered the inflow of fine particulate matter into the body and their harmfulness and analyzed changes in the concentration of indoor fine particulate matter according to the discharge method of a fixed air purifier and changes in the concentration of indoor fine particulate matter through the experiment, and the study evaluated the reduction performance of fine particulate matter in the breathing zone of occupants according to the operation method in which a movable air purifier responds to the movement of occupants.

## 2. Materials and Methods

### 2.1. Experiment Equipment and Method

The experiment was conducted indoor after sunset to minimize changes in indoor temperature caused by solar radiation. Figure 1 shows the experimental space and installation locations of the experiment’s equipment. The size of the experiment space is 7.0 m (L) × 10 m (W) × 2.8 m (H), and the volume is 196 m^3^. The experiment space was divided into four artificial zones, and the particulate matter (PM) concentration measuring devices were installed in the center of each zone. Pollutants were set to be generated from a height of 1.2 m from the center of the hallway wall. Table 1 shows the experiment devices used in this study. The target pollutant was a Polystyrene Latex (PSL) 2 μm standard solution, which was mixed with distilled water and sprayed in the space with the aerosol generator. The scanning electron microscope [21] was used to observe and analyze the particle shape of PSL, and the result showed the particle had an even spherical shape of 2.0 μm (SD ± 3%), as shown in Figure 2. The TES-5322 model was used for concentrating the PM concentration, and the measurement accuracy of the PM_2.5_ concentration is below ± 5μg/m^3^ when less than 50 μg/m^3^, and below ± 10% when greater than 50 μg/m^3^. The H13 grade filter that can filter out particles greater than 0.3 μm by 99.95% was applied.

The height of the PM concentration measuring device for measuring changes in the concentration of fine particulate matter in the breathing zone of occupants was set to 1.5 m by considering the breathing area of occupants while standing.

### 2.2. Arrangement of the Air Purifier Considering Draft

There are methods to control the concentration of various indoor pollutants such as fine particulate matter, including generation control, elimination control, and dilution control. If fine particulate matter is the target pollutant, the concentration can be controlled by eliminating the source or collecting fine particulate matter dispersed in the space. Therefore, the most effective method for reducing the concentration of indoor fine particulate matter is to locate and operate an air purifier close to the source of pollutants. However, outdoor fine particulate matter can flow in through openings or cracks in the building, and fine particulate matter are generated from a variety of activities by occupants, including cooking, exercising, and ventilation. Thus, the source of pollutants cannot be easily characterized. Furthermore, many measurement sensors are required to detect the source of the indoor fine particulate matter early [22,23,24,25]. This study selected a method of delivering air discharged from the air purifier toward the breathing zone of occupants to improve the reduction performance of the concentration fine particulate matter by operating the air purifier. In this case, occupants could inhale purified air regardless of the source. Contrarily, occupants could feel discomfort from drafts if the velocity of air currents is extremely high, and therefore, this needs to be considered. Gong et al. performed the study on the allowable wind velocity range needed for finding human recognition for the local airflow, under isothermal and non-isothermal conditions, and designed individual ventilation of the tropical regions through the experiment, and suggested that a wind velocity of minimum 0.3 m/s and maximum 0.9 m/s is acceptable based on the comfort of occupants [26].

This study set the installation height of the PM_2.5_ measuring device to 1.5 m by considering the breathing area of occupants while standing. In the case where purified air is discharged toward the breathing zone of occupants, the air purifier was arranged to maintain the velocity of the discharged air that reaches the 1.5 m-high measuring device at 0.8 m/s, as shown in Figure 3. The speed set here takes the precedence over the performance of reducing fine dust concentrations in the breathing zone and not over the human thermal comfort. The air purifier used in this study inhales air from the bottom and discharges from the top, and if the discharge angle is changed toward the breathing zone of the occupants, the velocity may change due to the pressure loss. Accordingly, a separate discharge outlet was made with a 3D printer to form the same face velocity regardless of the discharge angle, and it is shown in Figure 4.

### 2.3. Real-time Monitoring and Transfer Unit Production Using an Arduino Board

To reduce the concentration of fine particulate matter in the breathing zone of occupants through the operation of an air purifier, it is necessary to respond to the movement of occupants or monitor real-time changes of the concentration of fine particulate matter, as well as changes in the discharge angle of the air purifier. A data communication module for real-time monitoring of the measurement sensor for indoor fine particulate matter and a remote transfer unit for the air purifier were produced. Figure 5 shows the overview of the data communication module system, and Figure 6 shows the diagram of the remote transfer unit.

The PM_2.5_ concentration value monitored through the camera sensor is transferred to the server via ESP 32 WiFi module. Whether the air purifier needs to be moved is determined based on the PM_2.5_ concentration data received, and if its movement is necessary, the remote transfer unit sends a signal to the Arduino board to move it.

Figure 7 shows the data communication module and remote transfer unit for real-time monitoring of the movable air purifier made for the experiment of this study.

## 3. Measurement Experiment with Fixed Air Purifier

### 3.1. Case Setting

This experiment was conducted to evaluate the air purifiers’ performance in reducing the concentration of fine particulate matter in the breathing zone of occupants according to the location of the fixed air purifier and the direction of the discharged air. The air purifier was operated to make the background PM_2.5_ concentration of the experimental space to be less than 30 μg/m^3^. As shown in Figure 8, the experiment generated pollutants through the aerosol generator after 10 min from the start of measurement, and the air purifier was operated for 150 min through a remote control after 30 min from the operation of the aerosol generator. The generation of pollutants was discontinued after 90 min.

The case conditions according to the location of the air purifier and changes in the direction of discharge are shown in Table 2. In Case 1, the air purifier of upward discharge is located in the center, and it is the control group to compare with other cases in which the location and discharge direction of the air purifier were changed. In Case 2, the air purifier of the upward discharge was located on the nearby wall that was 1.4 m away from the P 4 (Point 4) measuring device and was set to analyze the reduction of the concentration of fine particulate matter in the breathing zone of occupants by installing the purifier near occupants regardless of the location of the pollutant source.

Case 3 and 4 were set to analyze the reduction of the fine particulate matter in local areas according to changes in the discharge direction of the air purifier. Case 3 set the direction of the discharge from the central location toward the P 4 measuring device as in Case 1, and the wind velocity of the air current reaching the P 4 measurement point was 0.23 m/s. Case 4 installed the air purifier near the occupants while changing the direction of the discharge, and the wind velocity of the air current reaching the P 4 measurement point was 0.81 m/s. The wind velocity value was the average value measured every 10 s for a total of 60 times with TSI 9565 (TSI Inc., Shoreview, MN, USA; Velocity).

### 3.2. Experiment Results

Figure 9 shows changes in the PM_2.5_ concentration according to the time by measurement point of each case. The changes in the concentration of fine particulate matter in the local areas according to the location of the air purifier were compared and analyzed with the results of Case 1 and Case 2. In all the measurement points of Case 1 and 2, the PM_2.5_ concentration has increased according to the pollutants, and the rising curve of the PM_2.5_ concentration was maintained consistently. When the generation of pollutants was discontinued, the PM_2.5_ concentration was decreased by the air purifier. In Case 1, the average PM_2.5_ concentration difference in P 1 (Point 1), P 2 (Point 2), P 3 (Point 3), and P 4 (Point 4) was about 25 μg/m^3^ from 40 min to 100 min. This is due to the distance from the source of pollutants to the measurement points. In Case 2, the PM_2.5_ concentration in P 4 that is close to the air purifier of the upward discharge was higher than the PM_2.5_ of the other measurement points. This was due to the movement of pollutants to the discharge outlet of the air purifier. In Case 3, P 4 also had the highest PM_2.5_ concentration and this meant that the P 4 point assumed as the breathing zone of the occupants did not fall within the influence of purified air discharged from the air purifier. The concentration of PM at points other than point P 4 was lower than that of P 4, but compared to Case 1, the PM concentration in P 3 decreased by 21 µg/m^3^, whereas points P 1 and P 2 maintained similar levels.

Meanwhile, the average PM_2.5_ concentration in P 4 from 40 min to 100 min was about 77 μg/m^3^ lower than other measurement points. In particular, the PM_2.5_ concentration in P 4 was maintained at less than 50 μg/m^3^ since the operation of the air purifier, and it reached 30 μg/m^3^ within 20 min after discontinuing the operation of the aerosol generator. It could be believed that the concentration of PM at point P 3 increased due to the diffusion of pollutants from the pollution source, and points P 1 and P 2 were similar to that of Case 1. Thus, the air purifier must be located near occupants, and the breathing zone of occupants must be located within the influence of purified air discharged from the air purifier to reduce the concentration of fine particulate matter in the breathing zone of occupants.

## 4. Measurement Experiment with Movable Air Purifier

### 4.1. Operating Method of Movable Air Purifier

In this section, the real-time monitoring device and the remote transfer unit were applied to allow the air purifier, which delivers purified air to the breathing zone of occupants, to respond to the movement of occupants and the reduction performance of fine particulate matter in the breathing zone of occupants was evaluated. In addition, the experiment was conducted through the real-time occupant tracking method and the zone controlling method by considering the threshold according to the moving path of the air purifier.

#### 4.1.1. Real-Time Occupant Tracking Method

Firstly, a controlling method of tracking occupants to reduce the concentration of fine particulate matter in the breathing zone of occupants, regardless of the surrounding concentration was set. The performance evaluation of this method was conducted through the measurement experiment.

The target experimental space and location for measuring the PM_2.5_ concentrations were the same as those in the experiment conditions of Section 2, but an additional device for measuring the concentration of fine particulate matter was installed at the P 5 (Point 5), which was an assumed breathing zone of the occupant. The measuring device for the concentration of fine particulate matter and the movable air purifier were fixed in the location, which was assumed as the breathing zone of the occupants. Figure 10 displays an implementation of a controlling method of tracking occupants in real-time. For the experiment, the aerosol generator and the air purifier were operated at the same time after 10 min from starting the recording of each measurement device, and the operation of the aerosol generator was discontinued after 190 min. The movement of the measuring device in the breathing zone of the occupants was set to move to random points every 15, 20, and 30 min to analyze the reduction effect of the concentration of fine particulate matter in the breathing zone of the occupants in terms of controlling the real-time tracking of occupants.

#### 4.1.2. Zone Controlling Method

Meanwhile, a method of controlling occupant tracking of a movable air purifier has limitations in its actual use. such as the moving path of the air purifier or changes in the number of occupants. Accordingly, the study conducted a measurement experiment for performance verification by setting the moving locations of the air purifier by zone, which allows the air purifier to move to the set location with the high concentration of fine particulate matter or the area where occupants are located and reduce the concentration of fine particulate matter in the breathing zone of occupants within that area. The experimental space and the locations of the device for measuring the concentration of fine particulate matter are the same as those in the measurement experiment for the occupant tracking method.

As shown in Figure 11, the experiment generated pollutants after 10 min from the start of measurement, and the air purifier was operated at the same time. The generation of pollutants was discontinued after 190 min. Figure 12 shows the moving path and location of the air purifier, and the air purifier was set to be located near the walls of each zone using the moving path of the wall.

The air purifier was relocated every 30 min to analyze changes in the concentration of indoor fine particulate matter, according to the moving path and location (Figure 13).

### 4.2. Experiment Results

Figure 14 shows changes in the PM_2.5_ concentration by measurement point according to the operation of the air purifier that tracks occupants in real-time.

The PM2.5 concentrations were compared and analyzed at P 5 (Point 5), which is considered as the breathing zone of occupants and continuously received purified air currents discharged from the air purifier and other measurement points. The PM_2.5_ concentrations at all measurement points had increased after 10 min from the start of the measurement and reached about 100 μg/m^3^ at P 1, P 2, and P 4 at 40 min. The upward slopes of P 5 and P 3, which were closest to the discharge outlet of the air purifier, were relatively low compared to those of other points and showed concentrations of 40 μg/m^3^ and 55 μg/m^3^, respectively. During the overall experiment time, the average concentration at P 5 was reduced by 51 μg/m^3^ compared to the average concentration at other measurement points, which demonstrated that the operation of an air purifier that tracks occupants in real-time could reduce the concentration of fine particulate matter in the breathing zone of occupants even if the concentration of fine particulate matter in the surrounding area is high. In addition, the concentration of fine particulate matter in measurement points near the discharge outlet was lower than other measurement points. This means the operation of an air purifier that tracks occupants in real-time could reduce the concentration of fine particulate matter in the breathing zone of occupants regardless of the surrounding concentration of fine particulate matter. The result of measuring the PM_2.5_ concentration at measurement points near the discharge outlet proves that the concentration of fine particulate matter could be reduced if the location is close to the air purifier and is within the influence of the purified air discharged from the air purifier.

Figure 15 shows changes in the PM_2.5_ concentration according to the time of each measurement point with the operation of the air purifier that controls zones through the set moving path.

The PM_2.5_ concentration in all the measurement points between 10–70 min located in the back of the room without moving an air purifier had increased to a similar level, reaching an average of 128 μg/m^3^ concentration at 70 min.

At 70–100 min, when the moved air purifier was located at the P 4 wall point, the average PM_2.5_ concentration at P 4 was about 56 μg/m^3^ for 30 min, and the PM_2.5_ concentration at P 1, P 2, and P 3 were 129, 133, and 172 μg/m^3^, respectively. The reason for the high PM_2.5_ concentration in P 3 was because pollutants generated from the aerosol generator were spread to the P 3 point due to the direction of the air current discharged from the air purifier. At 100–130 min, when the air purifier was located by the nearby wall of P 3 using the same moving path, a similar result for 70–100 min was observed. The average PM_2.5_ concentration in the P 3 point was about 60 μg/m^3^ for 30 min due to controlling of purified air discharged from the air purifier. P 4 showed the highest level of 161 μg/m^3^. At 130–160 min when the air purifier was located at the nearby wall of P 1, the PM_2.5_ concentration at P 1, which was affected by the purified air discharged from the air purifier, was about 56 μg/m^3^ for 30 min, and the PM_2.5_ concentrations at the P 2, P 3, and P 4 points were 153, 124, and 145 μg/m^3^ for 30 min, respectively. At 160–190 min, the PM_2.5_ concentration in P 3, which was controlled by purified air discharged from the air purifier, was 55 μg/m^3^ for 30 min, and the PM_2.5_ concentrations at P 1, P 2, and P 4 were 125, 129, and 151 μg/m^3^ for 30 min, respectively. The average concentration of the fine particulate matter from 70 min to 190 min in measurement points within the influence of purified air discharged from the air purifier was reduced by about 68 μg/m^3^ compared to other measurement points. This is due to the decreased removal rate of fine particulate matter in the breathing zone of the occupants for the corresponding time because the air purifier for controlling zones moves around.

## 5. Conclusions

This study performed measurement experiments to evaluate the reduction performance of fine particulate matter concentration in the breathing zone for a method of delivering purified air discharged from an air purifier to the human breathing zone. The results are as follows:

The method of installing a fixed air purifier at a location adjacent to occupants without changing the discharged direction cannot improve the performance in reducing the concentration of fine particulate matter within the breathing zone of occupants; the method of delivering purified air discharged from an air purifier can better reduce the concertation of fine particulate matter in the breathing zone of occupants compared to the concentration of fine particulate matter in the surrounding area, but if the distance from the air purifier to the controlling point is distant and so the velocity of airflow is not sufficient, there is no effect in reducing the fine particulate matter concentration.

In the case of a mobile air purifier, the real-time occupant tracking method was effective in terms of reducing the concentration of fine particulate matter in the breathing zone of occupants by 51 µg/m^3^ compared to the surrounding PM concentration, but there are limits in actual use regarding the moving path of the air purifier or the change in the number of occupants. On the contrary, the operation of the movable air purifier showed that the fine PM concentration of the occupant’s respiratory zone to be 68 µg/m^3^ lower than other measurement points. Thus, it is more effective to divide the target space by zone and move an air purifier around considering the number of occupants and the mobility of an air purifier.

This study compared and evaluated an air purifier’s performance in reducing the concentration of fine particulate matter in the breathing zone of occupants against measurement points of other zones by setting measurement points of each zone in one place when delivering purified air discharged from an air purifier. However, the level of reduction performance of fine particulate matter could differ according to the range of influencing purified air currents within the same zone. In addition, in the case of the zone controlling method, there are also limitations regarding the moving path and the distance between the occupants. Accordingly, a follow-up study on the range of purified air currents discharged from an air purifier, method of expanding the range, method of installing air purifier on the ceiling, and changing control mode according to the number of occupants will be conducted in future.

## Figures and Tables

**Figure 1 ijerph-17-05561-f001:**
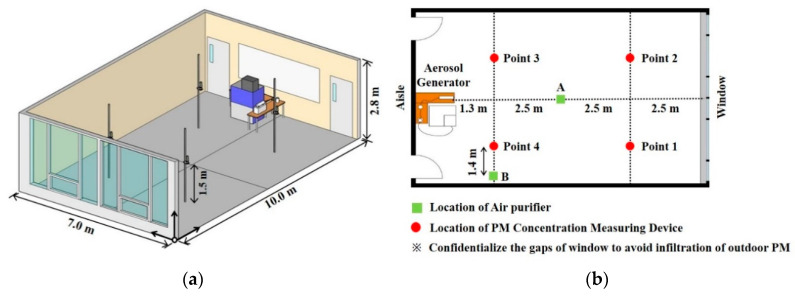
Experimental space and location of experiment equipment. (**a**) Experimental space; (**b**) Location of experiment equipment.

**Figure 2 ijerph-17-05561-f002:**
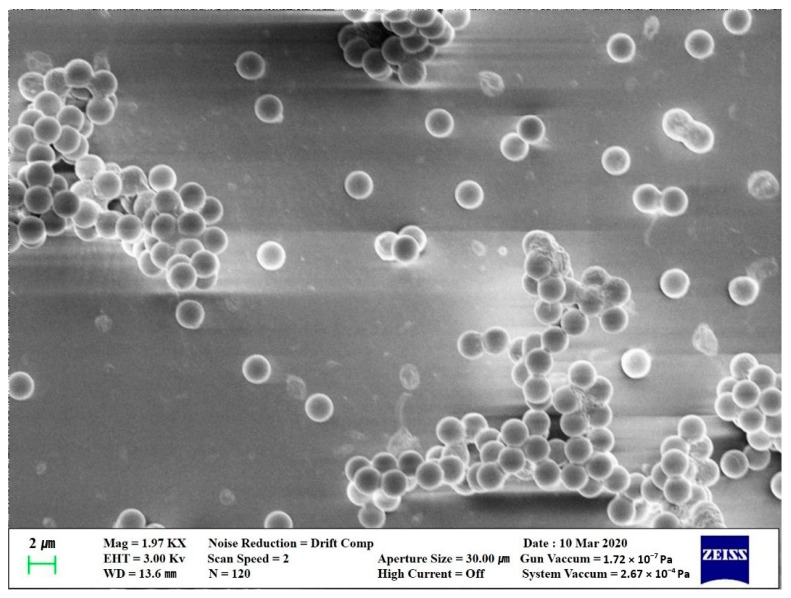
Expanded picture of standard solution through a scanning electron microscope (SEM).

**Figure 3 ijerph-17-05561-f003:**
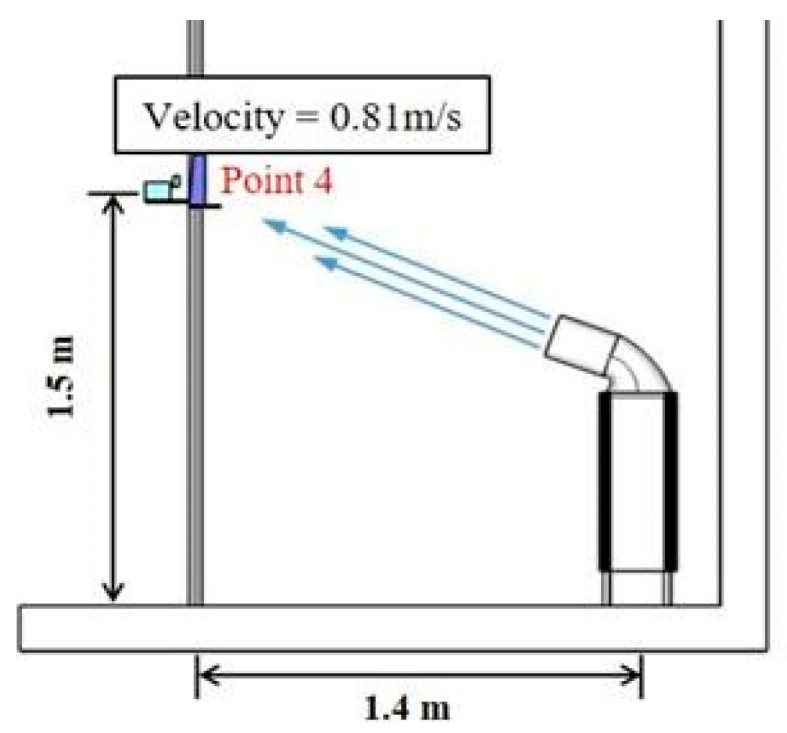
Air purifier of discharge direction toward breathing.

**Figure 4 ijerph-17-05561-f004:**
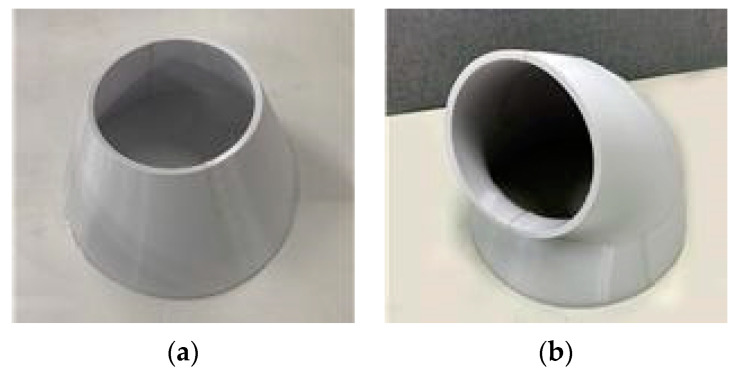
Discharge outlet made with a 3D print.: (**a**) Discharged upward (**b**) Discharged toward occupants.

**Figure 5 ijerph-17-05561-f005:**
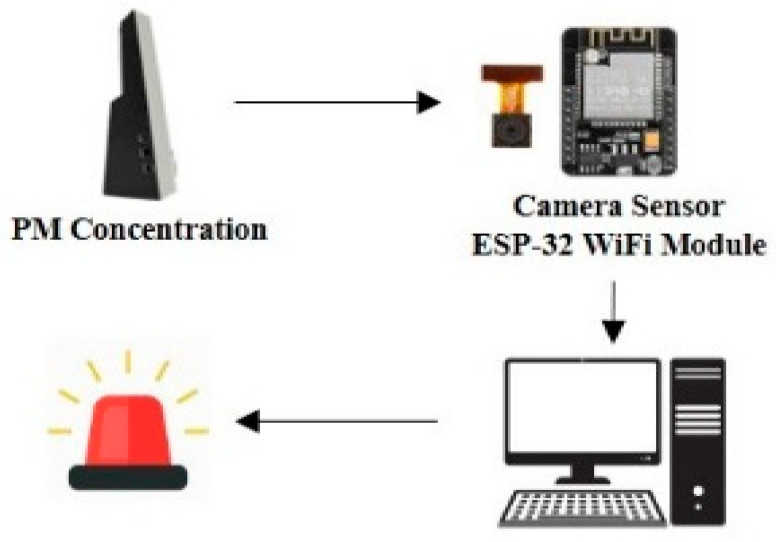
Overview of the data communication module system.

**Figure 6 ijerph-17-05561-f006:**
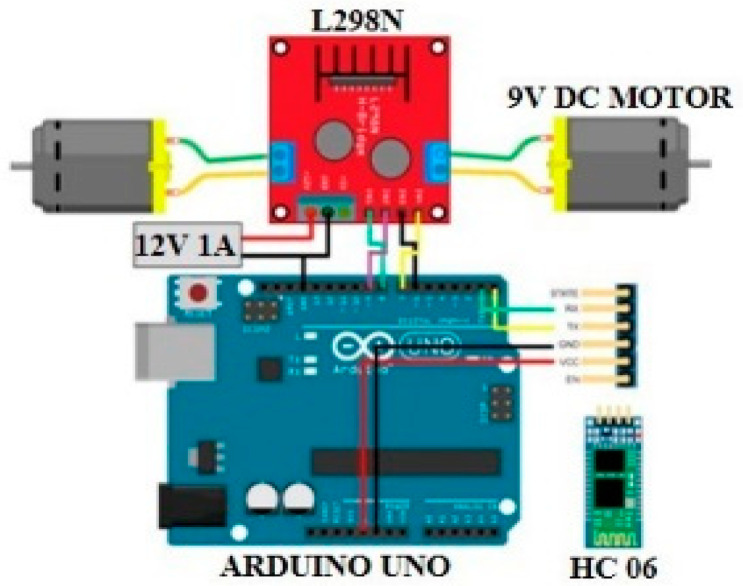
Diagram of the Arduino-based remote transfer unit.

**Figure 7 ijerph-17-05561-f007:**
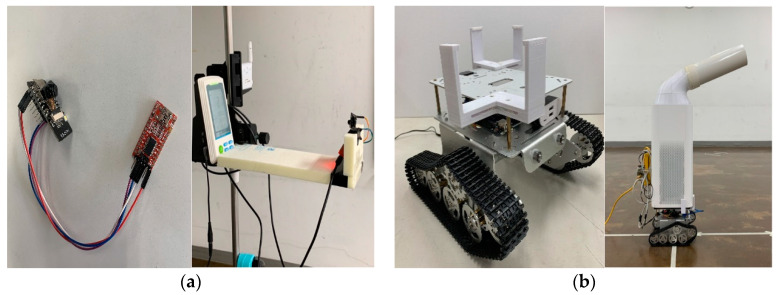
Data communication module and remote transfer unit used in the experiment: (**a**) Data communication module, (**b**) Remote transfer unit.

**Figure 8 ijerph-17-05561-f008:**
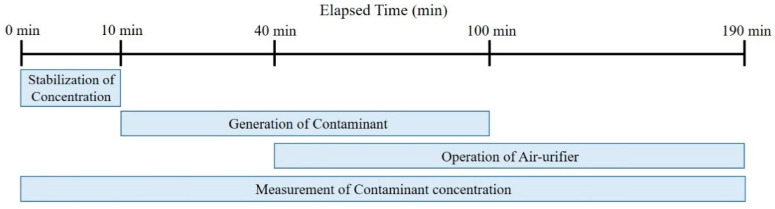
Experiment progress.

**Figure 9 ijerph-17-05561-f009:**
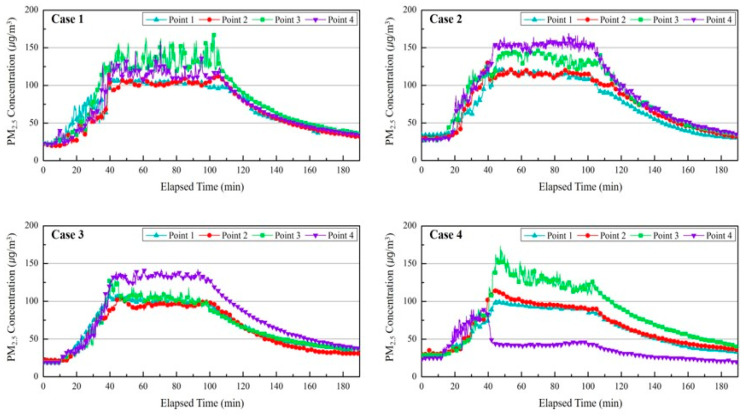
Comparing PM_2.5_ concentrations by measurement points (**Case 1–4**).

**Figure 10 ijerph-17-05561-f010:**
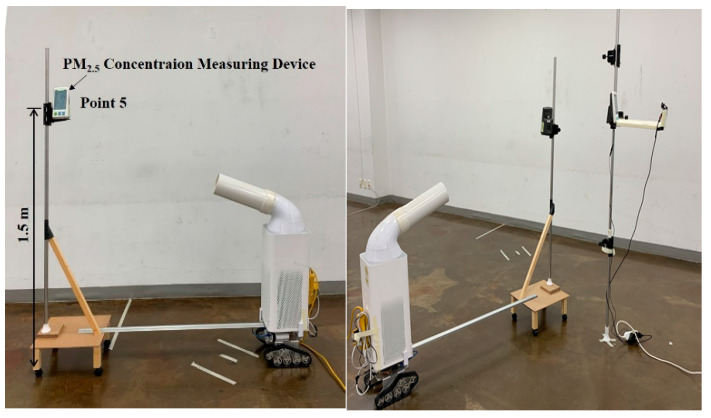
A method of implementing the air purifier that tracks occupants in real-time.

**Figure 11 ijerph-17-05561-f011:**
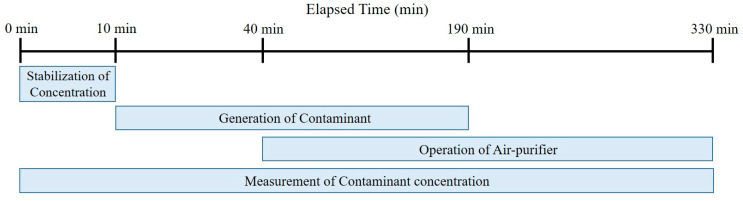
Experiment Progress (Real-Time Occupant Tracking Method and Zone Controlling Method).

**Figure 12 ijerph-17-05561-f012:**
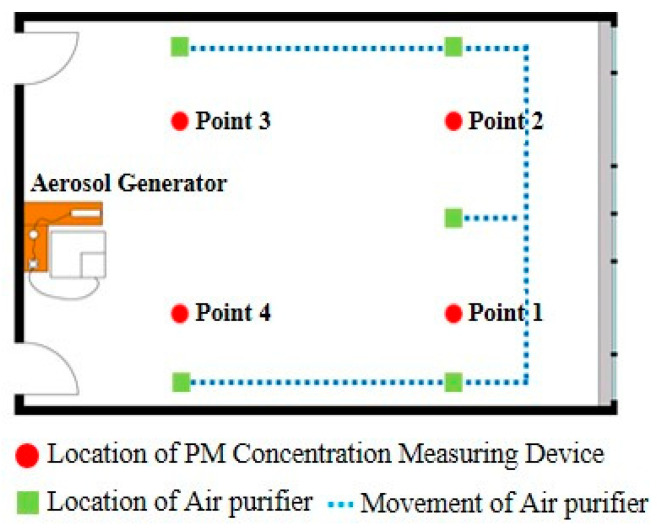
Moving path of the air purifier that controls zones.

**Figure 13 ijerph-17-05561-f013:**
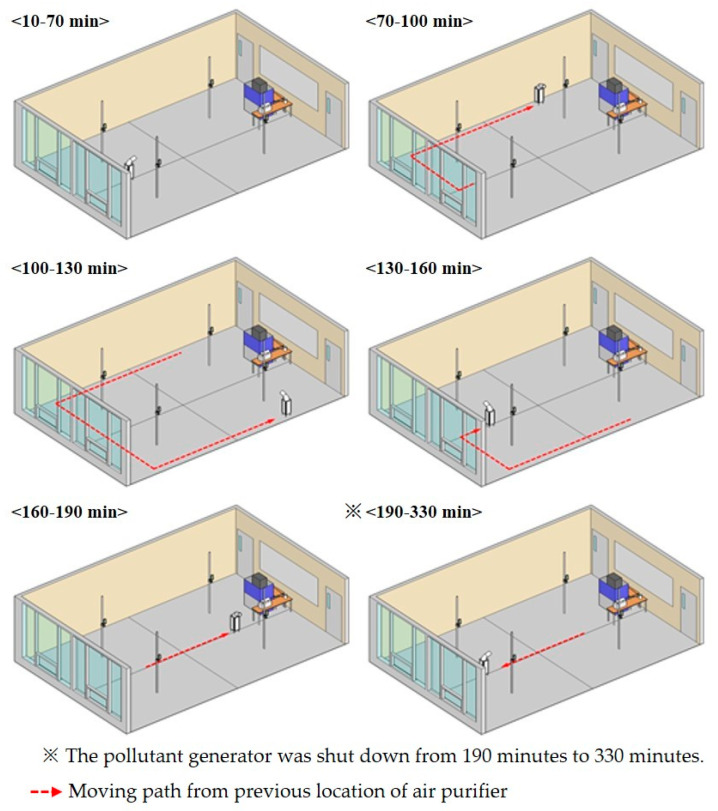
Location of the air purifier at each time and moving path from the previous location (Zone Controlling Method).

**Figure 14 ijerph-17-05561-f014:**
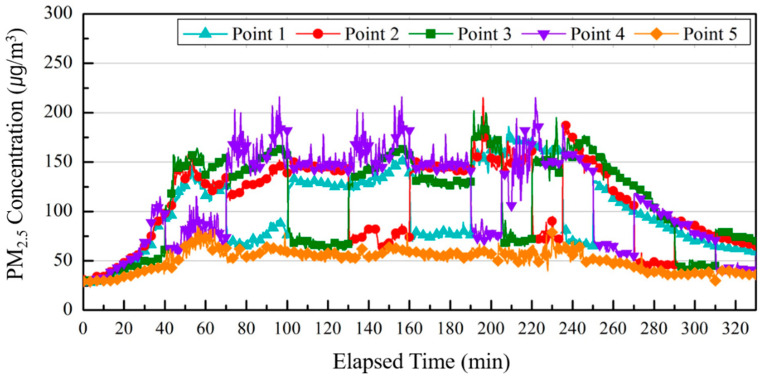
Comparing PM_2.5_ concentrations by measurement points according to the operation of the air purifier that tracks occupants.

**Figure 15 ijerph-17-05561-f015:**
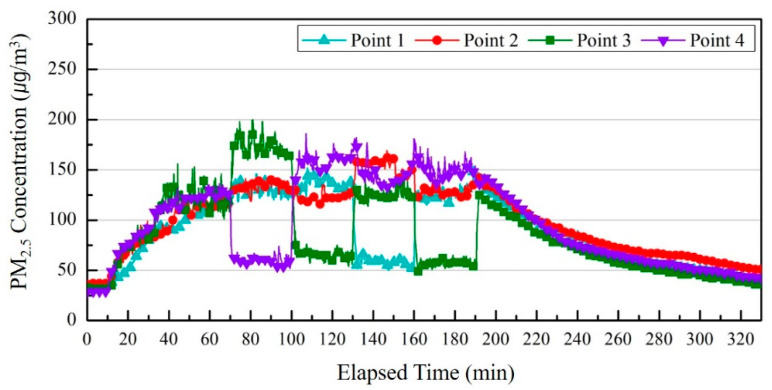
Comparing PM_2.5_ concentrations by measurement points according to the operation of the air purifier that controls zones.

**Table 1 ijerph-17-05561-t001:** Equipment and materials for experiment.

Purpose	Model	Quantity
Air purifier	Mi Air 2	1
PM measuring device	TES-5322	5
Thermo-hygrometer	TR-72WF	4
Anemometer	TSI 9565	1
PM_2.5_ generating device	Aerosol generator	1
Standard particle solution	Polystyrene Latex (Standard particle 15 mL)	3
Air supply and flow control of aerosol generator	Compressor (AM 400D)	1
Water removal of aerosol air supply	Dryer (TX15K)	1

**Table 2 ijerph-17-05561-t002:** Experiment case conditions.

Cases Variables	1	2	3	4	
Location of air purifier	A	B	A	B	
Direction of discharge	Upward		Toward point 4		
Initial PM concentration	27 μg/m^3^	26 μg/m^3^	29 μg/m^3^		30 μg/m^3^
Temperature/Relative Humidity	25 ± 2.2 °C/28 ± 1.8%				

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
