# Peer review of "Evaluation on Air Purifier’s Performance in Reducing the Concentration of Fine Particulate Matter for Occupants according to its Operation Methods"

_ijerph, 2020, doi:10.3390/ijerph17155561_

Round 1

Reviewer 1 Report

My detailed comments are as follows:

1 The literature review is inadequate. The number of related literatures listed in this manuscript is very small.

  1. The authors should quantitatively analyze the influence of various factors on the particle concentration and comfort of the occupants’ breathing zone. These factors include the distance between the purifier and people, the supply air speed, the location and intensity of the pollution source, etc.
  2. By directly supplying the purified air into the breathing zone, the airflow velocity in the breathing zone was 0.8 m/s in the experiment. People might feel uncomfortable under this airflow velocity; this airflow velocity is acceptable in the hot environments and unacceptable in the thermally neutral environments.

Reviewer 2 Report

The current manuscript presents an interesting approach in the evaluation of air purifiers. Instead of evaluating the improvement of indoor air quality (IAQ) in terms of fine particle (PM) concentration reduction, it applies a methodological approach which evaluate the purifier's performance in the breathing zone according to the operation method in which a movable purifier respond to the movement of occupants.

Authors describe well the experimental set up and the overall application seems to have significant effect in the IAQ in terms of fine PM reduction. However there are some issues which should be clarified:

1)The introduction should be enriched with more studies in the field of air purifiers. To the best of my knowledge there is a significant number of manuscripts dealing with the improvement of IAQ using air purifiers. Moreover is should be emphasized the different methodological approach applied in the current manuscript compared with other similar studies.

2) Although it is observed a significant decrease of fine PM around the breathing zone compared to the surrounding concentration it is really hard to understand whether this configuration could be applied and be effective in real world conditions. The set up of the experiment took place in a empty space specially designed for the current application. Wil all these settings be applicable in a real world environment e.g in an office space where a number of people will be working and moving around? I fully agree with the authors that such an application should be first testing in a "pilot mode". However I think that authors after setting and testing the application in an empty "test room" they should had applied the same methodology in a real world office (even in a small scale). Based on the methodology description, I am not convinced that the same results could be obtained.

Reviewer 4 Report

Overall, this study compared and evaluated the performance of an air purifier in reducing the concentration of fine particles in the occupants' breathing zone relative to measurement points in other zones. Two configurations were tested, (i) Measurement Experiment with Fixed Air Purifier; (ii) Measurement Experiment with movable Air Purifier

Main issues

Experience with a relatively complex methodology;

Unproven control of relevant variables;

Statistical treatment of the results nonexistent or rudimentar

In-depth review of the conclusions

In my opinion, a thorough review of the methodology followed in the experiment should be made, as some relevant procedures are missing (or have not been followed and therefore are not presented or explained).

Uncontrolled variables should be clearly identified, such as the mixture caused by the movement of people and which in this experiment is omitted.

More than one sensor is used without any performance assessment procedure being mentioned - intercomparison of results and study of possible reading deviations between the sensors used, which may have conditioned the observed results and the conclusions drawn.

The data processing has to be reviewed and statistical methods of comparison of series should be applied in order to understand whether or not there are statistically significant differences between the observed series.

The conclusions have to be reviewed, as they do not reflect the results of the work in a correct way and the lines of future research are not clearly presented and defined.

Minor issues

  1. Line 37 (...) high levels of fine particulate matter are formed (...)

in fact, this statement is not completely true, since, besides the formation of secondary particles in the atmosphere, we also have the primary emission of particles that may also be relevant, depending on the regions;

  1. Line 42 and 43 (...) Fine particulate matter are classified into PM10 which has particulate matter less than 10µm and PM2.5 which has particulate matter less than 5 µm (...) - Please review. This sentence contains several scientific inaccuracies;
  2. Line 49 (...) In particular, ultrafine particulate matter less than 2.5 µm(…) - Ultra-fine particles are particles smaller than 0,1µm. Please review the sentence...
  3. Lines 73 – 74 (...) The experiment was conducted indoor after sunset, and the blinds were installed on the windows to minimize the influence of solar radiation. (...) - this sentence is incoherent. Please review the sentence. It is relevant to explain why you want an environment with a low level of solar radiation. Please, also provide the appropriate explanation supported by a reference.
  4. In Figure 8 correct "operation of air-purifier"
  5. The authors should consider placing Figure 1 b) in section 3.1 to support the experience description.

Round 2

Reviewer 4 Report

I would like to thank the authors for their review. In fact, in my opinion, the manuscript is clearer. I found some small details that can easily be corrected.

Regarding the answers given, I have the following comments:

Comment 1 - Okay;
Comment 2 - Okay;
Comment 3 - The answer given, in particular the comparison with the AM510 TSI sensor, only makes me more concerned. In fact, even for sensors calibrated by the manufacturer, when we have a simultaneous use, as is the case, it is good practice to make a previous intercomparison of the sensors performance. This procedure corresponds to putting all the sensors to measure at the same time and subject to the same conditions, in a sequential way for several situations of controlled environment. This procedure should be followed by a robust statistical treatment that confirms (or not) any existing deviations between the different sensors in identical controlled environments.
Comment 4 - I am not comfortable with the answer. In fact, if there are several series of values observed under different conditions, the comparison between the series being presented as a simple percentage variation is not a good practice and is scientifically unsound. The difference you observed is statistically significant or not? with what level of confidence?
Comment 5 - I am already more comfortable with the conclusions. I maintain a challenge for the authors. I know that the whole experiment was conducted if the presence of any occupant. However, the idea is to prove that the system can be more effective in a real application. What can happen when the system is used effectively with occupants moving within space causing localized turbulence? will this system, tested under very controlled conditions, remain efficient?

Still some minor problems:

Line 41: Please update the mortality figure presented and the corresponding reference. There are new WHO estimates.

Line 51: (…) fatal to the body (…); Please, revise

Line 61: (…) We thought (…); Please, change to (…) we thought (…)

Table 1 (…) Thermo-hygrometer             TR-72WF             ea (…);Please, change to (…)Thermo-hygrometer                TR-72WF                 1ea (…)
